# Implementation of Machine Learning Software on the Radiology Worklist Decreases Scan View Delay for the Detection of Intracranial Hemorrhage on CT

**DOI:** 10.3390/brainsci11070832

**Published:** 2021-06-23

**Authors:** Daniel Ginat

**Affiliations:** Department of Radiology, University of Chicago, Chicago, IL 60615, USA; dginat@radiology.bsd.uchicago.edu

**Keywords:** head, CT, hemorrhage, artificial intelligence, report, scan view delay

## Abstract

Background and Purpose: Prompt identification of acute intracranial hemorrhage on CT is important. The goal of this study was to assess the impact of artificial intelligence software for prioritizing positive cases. Materials and Methods: Cases analyzed by Aidoc (Tel Aviv, Israel) software for triaging acute intracranial hemorrhage cases on non-contrast head CT were retrospectively reviewed. The scan view delay time was calculated as the difference between the time the study was completed on PACS and the time the study was first opened by a radiologist. The scan view delay was stratified by scan location, including emergency, inpatient, and outpatient. The scan view delay times for cases flagged as positive by the software were compared to those that were not flagged. Results: A total of 8723 scans were assessed by the software, including 6894 cases that were not flagged and 1829 cases that were flagged as positive. Although there was no statistically significant difference in the scan view time for emergency cases, there was a significantly lower scan view time for positive outpatient and inpatient cases flagged by the software versus negative cases, with a reduction of 604 min on average, 90% in the scan view delay (*p*-value < 0.0001) for outpatients, and a reduction of 38 min on average, and 10% in the scan view delay (*p*-value <= 0.01) for inpatients. Conclusion: The use of artificial intelligence triage software for acute intracranial hemorrhage on head CT scans is associated with a significantly shorter scan view delay for cases flagged as positive than cases not flagged among outpatients and inpatients at an academic medical center.

## 1. Introduction

Intracranial hemorrhage is a common occurrence, with an estimated 37,000 to 50,000 cases in the United States annually [1]. Half of the resulting mortality occurs within the first 24 h and early treatment can improve outcomes [2,3]. In particular, hematoma expansion can lead to neurologic decline and irreversible damage can result as early as within the first few hours after onset, making prompt and accurate diagnosis via neuroimaging essential for appropriate management of these patients [1]. However, a high scan volume can be a burden on radiologists and scans can remain on a queue for a prolonged period of time until they are interpreted [4].

Artificial intelligence software can be used to identify urgent abnormalities on head CT scans and automate the triage process for scan interpretation [5]. For example, the Aidoc (Tel Aviv, Israel) software has a reported sensitivity of 89 to 95% and a specificity of 94 to 99% for the detection of acute intracranial hemorrhage on CT [6,7]. In addition, radiologists can be alerted of urgent cases in real time via flags and widgets integrated into the PACS workstation. Ultimately, the intent of such software features is to expedite communication of urgent findings.

The purpose of this study was to assess the impact of a commercial artificial intelligence prioritization tool from Aidoc on scan view delay time for any acute intracranial hemorrhage depicted on CT. In particular, the hypothesis of this study is that the head CT scans flagged as positive by the artificial intelligence system have a shorter scan view delay than cases that are not flagged, especially for outpatient cases.

## 2. Materials and Methods

### 2.1. Artificial Intelligence System

The deep learning triage software developed by Aidoc (Tel Aviv, Israel) was utilized. The software is based on proprietary convolutional neural network architecture. The development dataset included approximately 50,000 CT studies collected from 9 different sites. In total, data was derived from 17 different scanner models. CT data slice thicknesses ranged from 0.5 to 5 mm. Data from all anatomic planes were used when available. Only soft-tissue kernel images were used. Ground truth labeling structure varied depending on hemorrhage type and size, and included both weak and strong labeling schema. Label types included study-level classification for diffuse subarachnoid hemorrhage, slice-level bounding boxes around indistinct extra- and intra-axial hemorrhage foci, and pixel-level semantic segmentation of well-defined intraparenchymal hemorrhage [6]. The algorithm is not tuned to any hemorrhage threshold size. The average processing time for intracranial hemorrhage, defined as the time between the data reaching the Aidoc server to the result being available to the radiologist on PACS, is under 3 min. The software can be connected in a variety of manners to PACS and all relevant scans are automatically sent for analysis with no need for a manual trigger by the radiologists. The software is vendor neutral and is FDA cleared for use on multiple scanners from multiple manufacturers.

### 2.2. Software Integration and Deployment

The reported ground truth labeling is related to the model training, validation, and verification data conducted by the manufacturer of the software and is not related to the current study. However, for the current study, the final radiologist report defined the ground truth. All urgent, non-contrast head CT scans performed at a single academic medical center were automatically and immediately forwarded for analysis by the software in real-time. The scans were performed on 9 different scanners, each using 120 kVp, but with variable tube currents ranging from 185 to 350 mA, and axial sections with 5 mm slice thickness, but coronal and sagittal sections with 3 mm slice thickness. The scans were sent to analysis with no additional data or clinical context of the study. Once a positive ICH case is detected, the software sends a notification through a standalone application and also flags and elevates the case in the worklist. In addition, the Aidoc software includes a widget that lists studies with positive findings and flashes on the workstation screen as soon as a new positive case appears on the worklist (Figure 1). The widget could be installed on any workstation within the institution. There were 5 participating radiologists who were trained and instructed on the implementation of the software. The participating radiologists were a subset of the department radiologists that read scans from the locations included in the analysis, but all were the department radiologists with active widgets.

### 2.3. Dataset

All adult non-contrast head CT scans between May 2020 to February 2021 at a single tertiary care academic institution with a level 1 trauma center that were evaluated for acute intracranial hemorrhage by the software were included in this assessment. Approval for this study was obtained from the Institutional Review Board and informed consent was waived.

### 2.4. Analysis

The scan view delay time was calculated as the difference between the time the study was completed on PACS and the time the study was first opened for reporting by one of the participating radiologists. The scan view delay was stratified by scan location, including emergency, inpatient, outpatient, and otherwise unspecified. The mean scan view delay times for cases flagged as positive by the software were compared to those that were not flagged using two-tailed *t*-tests. A *p*-value of less than 0.05 was considered statistically significant. The sensitivity, specificity, positive predictive value, and negative predictive value for the Aidoc software for acute intracranial hemorrhage detection during the timeframe of this study were also determined.

## 3. Results

A total of 8723 head CT scans were assessed by the software for acute intracranial hemorrhage, including 6894 cases that were not flagged and 1829 cases that were flagged as positive, the majority of which were ED (41.3%) and inpatient cases (41.1%). The scan view delay was lower for flagged cases than non-flagged cases for all scan locations, including a difference of 12.5 min (−14.8%) for emergency cases, 37.5 min (−9.6%) for inpatients cases, 57.9 min (−34.7%) for other (unclassified patient class), and 603.9 min (−89.6%) for outpatients (Table 1). However, the difference was statistically significant for the outpatient and inpatient cases (*p*-value < 0.001, and *p* = 0.002), but not the emergency (*p*-value = 0.37) or other (*p*-value = 0.25) cases. The boxplots of the scan view delay time for the positive and negative cases are shown in Figure 2. The sensitivity, specificity, positive predictive value, and negative predictive value for the Aidoc software for acute intracranial hemorrhage detection during the timeframe of this study are listed in Table 2. For emergency cases, the time reduction was most prominent during the 9 p.m. to 3 a.m. and 10 a.m. to 12 p.m. periods, especially during the weekend. However, for inpatients and outpatients, the reduction was independent of day or time of day.

## 4. Discussion

Machine learning can effectively detect intracranial hemorrhage on CT and can reduce the false negative rate for the detection of intracranial hemorrhage [6,7,8,9,10,11]. Consequently, artificial intelligence software has begun to be incorporated into the clinical workflow. This study shows that the Aidoc deep learning software for the detection and prioritization of acute intracranial hemorrhage on head CT scans is associated with a significantly lower scan view delay for outpatient and inpatient cases in a diverse clinical setting.

Artificial intelligence can also have an impact on other aspects of the radiology workflow. For example, a convolutional neural network developed to screen head CT scans for acute neurologic events was found to preprocess images, run its inference method, and potentially raise an alarm 150 times faster than humans in a simulated clinical environment [10]. Similarly, deep learning architecture software reportedly reduced the time to diagnosis of new intracranial hemorrhage on outpatient CT scans by 96% [11]. Furthermore, the use of artificial intelligence led to a reduction in report turnaround time in a level 1 trauma center [12].

Notably, the impact of the artificial intelligence software was not the same across all scan settings. While there was a trend towards a lower scan view delay for flagged emergency cases, the reduction was not statistically significant. This is likely related to the inherently fast-paced workflow for emergency cases regardless of the findings, with less of a margin for further expediting flagged cases. In particular, the overnight emergency radiologists only focus on emergency cases and occasionally STAT inpatients, while the daytime neuroradiologists interpret emergency, inpatient, and outpatient cases concurrently. 

Although not statistically significant, the lower scan view delay for flagged emergency cases in this study could potentially be clinically significant in some instances. The scan view time for outpatient scans is generally longer than for emergency and inpatient scans due to an existing prioritization hierarchy at our institution, whereby the radiologists attend to emergency and inpatient scans more promptly, particularly if those are designated as STAT on the worklist. However, it is possible that radiologists decide that flagged cases are not particularly urgent and might postpone reporting such cases if the hemorrhages are small or if these are false positives, such as artifacts, based on the selected images provided by the widget, although these were not frequently observed given the positive predictive value of 85.9%. Indeed, the ability to access selected images on the widget while reviewing a scan facilitates an efficient workflow to determine if a notification was true or false positive.

Ultimately, the influence that the artificial intelligence prioritization software can have is dependent upon the sensitivity and specificity of the software, which were similar in this study to prior reports [6,7], and the radiologist’s response to the alerts from the software. For example, some radiologists might choose to finish reporting a case rather than immediately move on to a newly flagged case, particularly if the current case is designated as STAT and also has potentially urgent findings. 

Various other factors can affect scan view delay, including case volume, individual radiologist experience or subspecialization, work habits, such as batch reading, and interruptions from noninterpretive tasks, such as phone calls [13,14,15]. Indeed, a wide range of scan view delay differences for flagged and non-flagged cases among various radiologists was observed in this study. Notably, the longer scan view delay for flagged cases for some of the radiologists in the study may have been attributable to a disproportionate number of head CT scans combined with other studies, such as with trauma series, which made the overall review more time consuming.

A limitation of this study is that there was a small, but heterogeneous group of participating radiologists, including both attending neuroradiologists and overnight emergency radiologists, each of whom could have different work habits. Conversely, the study was conducted at a single academic medical center, whereby the findings of this study might not translate to other types of practice settings. In addition, while decreased scan view delay might be expected to have a positive impact on patient case, the clinical impact was not specifically assessed. Finally, a portion of scans could not be classified in terms of scan location and were not included in further analysis.

## 5. Conclusions

The use of artificial intelligence triage software for acute intracranial hemorrhage on head CT scans is associated with a significantly shorter scan view delay for cases flagged as positive than cases not flagged among outpatients and inpatients at an academic medical center, but not for emergency department cases.

## Figures and Tables

**Figure 1 brainsci-11-00832-f001:**
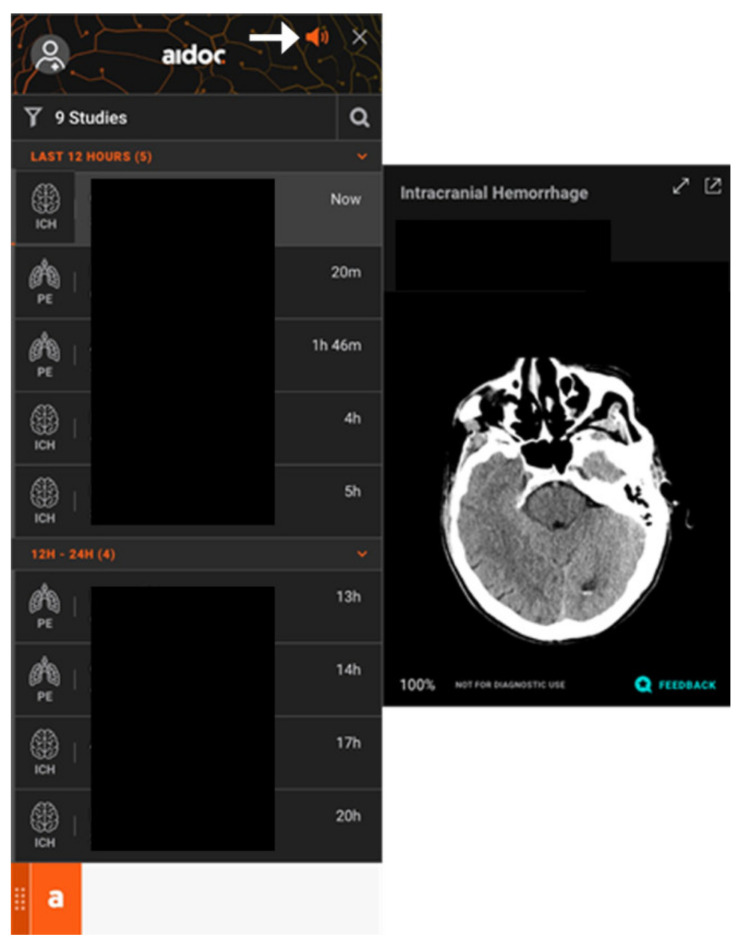
Example of the Aidoc widget on the workstation that alerts the radiologist when new flagged cases appear on the PACS worklist. The widget lists studies that are positive for acute intracranial hemorrhage and selected images with findings can be viewed for each patient. The widget includes an audible alert feature (arrow). The patient names on this image are hidden for anonymity.

**Figure 2 brainsci-11-00832-f002:**
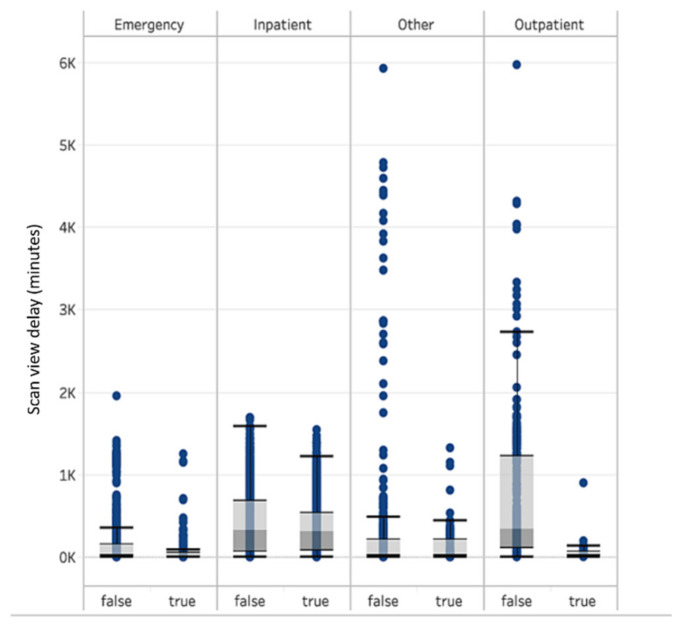
Boxplots of the scan view delay for flagged and non-flagged cases.

**Table 1 brainsci-11-00832-t001:** Scan view delay times for different patient locations.

	Emergency		Inpatient		Other		Outpatient	
	Negative	Positive	Negative	Positive	Negative	Positive	Negative	Positive
Number of scans	3398	201	2130	1457	987	132	379	39
Median scan view time delay (minutes)	16	14	298	298	14	18	326	35
Average scan view time delay (minutes)	85	72	390	352	167	109	674	70
Standard deviation (minutes)	194	177	368	315	572	217	825	141

**Table 2 brainsci-11-00832-t002:** Performance parameters of the Aidoc software for acute intracranial hemorrhage detection.

Sensitivity	88.4% (95% confidence interval: 87.88% to 89.82%)
Specificity	96.1% (95% confidence interval: 95.65% to 96.52%)
Positive Predictive Value	85.9% (95% confidence interval: 84.36% to 87.34%)
Negative Predictive Value	96.3% (95% confidence interval: 95.86% to 96.71%)

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
