# Peer review of "Implementation of Machine Learning Software on the Radiology Worklist Decreases Scan View Delay for the Detection of Intracranial Hemorrhage on CT"

_brainsci, 2021, doi:10.3390/brainsci11070832_

Round 1

Reviewer 1 Report

This is a well written, clear and concise analysis of the efficiency improvement after implementation of an artificial intelligence decision support system. In particular, the choice to focus on intracranial hemorrhage is relevant as rapid diagnosis is of utmost clinical importance in relation to this finding. Furthermore, the topic is timely, as artificial intelligence research in Radiology has focused primarily on development and accuracy of algorithms and rarely meaningfully quantifies increases in efficiency and ultimately patient outcomes.

One would expect a small or negligible difference in report turnaround for emergency room patients as this list is constantly and rapidly being assessed by the workforce. The research confirms this intuition. Consequently, one would expect the strongest value of the algorithm in relation to outpatient populations (lists that are less efficiently scrutinized).  The manuscript quantifies this and the results are remarkable:  a reduction of 604 minutes on average (90%) in the scan view delay for outpatients.

Abstract and Introduction: Interesting, clear and well written sections.

Material and Methods: In the Artificial intelligence system section, the authors state that “Ground truth labeling structure varied depending on hemorrhage type and size, and included both weak and strong labeling schema.”  My presumption is that this statement refers to the validation of the algorithm prior to implementation at the institution.  If so, it is unclear from the Software integration and deployment section how ground truth was established for the basis of this study. Please clarify how ground truth was established in relation to the data used in this study.

Please specify what the algorithm processing time is on average.  In other words, what is the algorithms inherent processing delay?

Results: The authors note a Sensitivity of 88.4%. Was any false negative analysis performed?  What can we learn from these cases? What was the difference in delay between true positives and false negatives? A work prioritization tool should ideally have the highest sensitivity possible in order to prioritize the majority of relevant cases. Please comment in the Discussion section the implications of this sensitivity level.

The authors note a Specificity of 96.1% which is exemplary. However, in my experience, algorithms often flag hyperdense pathology such as meningiomas or cavernous malformations. Please comment as to the efficiency implications of prioritization of false positives.

Was any analysis conducted in relation to the clinical relevance (acuity) of the findings?  For example, reduced delays in regards to an unchanged outpatient follow up subdural hemorrhage is of no clinical significance, while reduced delays in regards to a new, unexpected outpatient intracranial hemorrhage is of utmost importance.  Please comment.

Discussion:  Well written and informative. Please comment on the issues noted in the Results feedback.

Author Response

Material and Methods: In the Artificial intelligence system section, the authors state that “Ground truth labeling structure varied depending on hemorrhage type and size, and included both weak and strong labeling schema.”  My presumption is that this statement refers to the validation of the algorithm prior to implementation at the institution.  If so, it is unclear from the Software integration and deployment section how ground truth was established for the basis of this study. Please clarify how ground truth was established in relation to the data used in this study.

The reported ground truth labeling is related to the model training, validation and verification data conducted by the manufacturer of the AI solution and not related to the current study.

For the current study the final radiologist report was taken to define the ground truth.

Please specify what the algorithm processing time is on average.  In other words, what is the algorithms inherent processing delay?

The average processing time for intracranial hemorrhage, defined as the time between the data reached the Aidoc server to the result being available to the radiologist on PACS, is under 3 minutes.

Results: The authors note a Sensitivity of 88.4%. Was any false negative analysis performed?  What can we learn from these cases? What was the difference in delay between true positives and false negatives? A work prioritization tool should ideally have the highest sensitivity possible in order to prioritize the majority of relevant cases. Please comment in the Discussion section the implications of this sensitivity level.

The study did not explore this. This will be taken into consideration for future investigation.

The authors note a Specificity of 96.1% which is exemplary. However, in my experience, algorithms often flag hyperdense pathology such as meningiomas or cavernous malformations. Please comment as to the efficiency implications of prioritization of false positives.

In the solution performance evaluation we report a PPV of 85.9% which demonstrates that FPs were not frequently observed. Furthermore, using the widget, integrated in the reading workflow, allowing access to selected images whilst reviewing the study enabled a very efficient workflow to determine if a notification was TP or FP.

Was any analysis conducted in relation to the clinical relevance (acuity) of the findings? For example, reduced delays in regards to an unchanged outpatient follow up subdural hemorrhage is of no clinical significance, while reduced delays in regards to a new, unexpected outpatient intracranial hemorrhage is of utmost importance. Please comment.

The algorithm identifies acute ICH. While ICH is generally deemed of clinical significance, the study did not specifically explore this. This will be considered for future investigation.

Reviewer 2 Report

This is a well-conducted study with high potential for change in practice.  A few points are worthwhile of further discussion:

  1. It would be interesting to see more of the bleed characterizations for each of the bins they reported on (location and size for flagged versus not flagged).  The author mentions in the discussion that the hemorrhages may be small in the outpatient setting flagged cases leading to the delay between reads since they are not clinically pressing.  Seems to be an easy thing to actually back with volumes or other bleed subgrouping tags and gives more weight to the conclusion that they can reduce read delays in the outpatient setting with some clinical context. If the largest bleed they “reduced the scan read delay” for is under 5 ccs or secondary to artifact, the reduction reported of 10 hours may not be clinically significant for this setting.
  2. Similarly, what was the threshold used to deem a scan “positive” to be flagged.  Although any intracranial hemorrhage including SAH and extra-axial hemorrhages were included, is the algorithm tuned to a certain size.
  3. Classifications of bleeds missed is also important, a sensitivity of 88% is pretty low when talking about ICH identification without knowing the size threshold (see comment 2). Table 2 reproduced by scanner location would be a nice addition to see the sens/spec in the outpatient setting where the time savings is reported in particular.
  4. It is unclear how the scans were pushed to the AI software for reads.  Is it embedded within the scanners themselves and linked to the PACS system? Can this widget be installed on any software or is it specific to the types of scanners and systems at this one center?
  5. All scans were taken from May 2020 to February 2021 but it is not clear whether the 5 participating radiologists comprised the entirety of the radiology department and were assigned to read for all locations included in this analysis.
  6. Did the scan read delay differ by time or type of day (day shift/night shift; start vs. end of shift;  weekday vs. weekend)?  This may be useful for determining when the delays can be most impacted.
  7. In Figure 2. Boxplots of the scan view delay for flagged and non-flagged cases, would it be more accurate to label the x-axis as flagged and non-flagged compared to "true" and "false" or what is the meaning of true and false.
  8.  Are statistical comparisons of scan view delay based on mean or median values;  were these normally distributed?

Author Response

  1. It would be interesting to see more of the bleed characterizations for each of the bins they reported on (location and size for flagged versus not flagged).  The author mentions in the discussion that the hemorrhages may be small in the outpatient setting flagged cases leading to the delay between reads since they are not clinically pressing.  Seems to be an easy thing to actually back with volumes or other bleed subgrouping tags and gives more weight to the conclusion that they can reduce read delays in the outpatient setting with some clinical context. If the largest bleed they “reduced the scan read delay” for is under 5 ccs or secondary to artifact, the reduction reported of 10 hours may not be clinically significant for this setting.

The solution does not provide bleed volume measurements and adding this by hand would be beyond the scope of the study. This is an interesting question for future research.

  1. Similarly, what was the threshold used to deem a scan “positive” to be flagged.  Although any intracranial hemorrhage including SAH and extra-axial

hemorrhages were included, is the algorithm tuned to a certain size.

The algorithm is not tuned to any hemorrhage threshold size.

  1. Classifications of bleeds missed is also important, a sensitivity of 88% is pretty low when talking about ICH identification without knowing the size threshold (see comment 2). Table 2 reproduced by scanner location would be a nice addition to see the sens/spec in the outpatient setting where the time savings is reported in particular.

This data is unavailable.

  1. It is unclear how the scans were pushed to the AI software for reads.  Is it embedded within the scanners themselves and linked to the PACS system? Can this widget be installed on any software or is it specific to the types of scanners and systems at this one center?

The AI software can be connected in a variety of manners, ie: PACS, directly to modalities or via DICOM routers. All relevant scans are being automatically sent to analysis with no need for a manual trigger by the radiologists. The widget can be installed on any workstation within the institution. The AI software is vendor neutral and is FDA cleared for GE, Siemens, Philips and Cannon CT scanners.

  1. All scans were taken from May 2020 to February 2021 but it is not clear whether the 5 participating radiologists comprised the entirety of the radiology department and were assigned to read for all locations included in this analysis.

The participating radiologists were a subset of the department radiologists to read scans from the locations included in the analysis, since they were they only ones to have the widget installed.

  1. Did the scan read delay differ by time or type of day (day shift/night shift; start vs. end of shift;  weekday vs. weekend)?  This may be useful for determining when the delays can be most impacted.

We do see that for Emergency cases - the reduction of wait time is most prominent during 9 pm - 3 am and 10 am-12 pm, and especially during the weekend. For Inpatients and Outpatients the reduction is independent of days / time of day.

  1. In Figure 2. Boxplots of the scan view delay for flagged and non-flagged cases, would it be more accurate to label the x-axis as flagged and non-flagged compared to "true" and "false" or what is the meaning of true and false.

We agree with the reviewer and have adjusted the figure to reflect the positive/negative terminology used throughout the rest of the paper.

  1. Are statistical comparisons of scan view delay based on mean or median values;  were these normally distributed?

The statistical comparisons were performed using the mean values.

Round 2

Reviewer 2 Report

Thank you for appropriate revisions.  All of my questions were answered.